# Urban Vegetation Classification for Unmanned Aerial Vehicle Remote Sensing Combining Feature Engineering and Improved DeepLabV3+

Qianyang Cao [1,2,†] ![ID], Man Li [1,2,†], Guangbin Yang [1,2,*], Qian Tao [1,2], Yaopei Luo [1,2], Renru Wang [1,2] and Panfang Chen [3]

1    School of Geography and Environmental Science, Guizhou Normal University, Guiyang 550025, China; caoqianyang_gz@163.com (Q.C.); liman4515@163.com (M.L.)
2    Guizhou Key Laboratory of Mountain Resources and Environment Remote Sensing, Guiyang 550025, China
3    Guizhou First Surveying and Mapping Institute, Guiyang 550025, China
*    Correspondence: ygbyln@163.com
†    These authors contributed equally to this work.

**Abstract:** Addressing the problems of misclassification and omissions in urban vegetation fine classification from current remote sensing classification methods, this research proposes an intelligent urban vegetation classification method that combines feature engineering and improved DeepLabV3+ based on unmanned aerial vehicle visible spectrum images. The method constructs feature engineering under the ReliefF algorithm to increase the number of features in the samples, enabling the deep learning model to learn more detailed information about the vegetation. Moreover, the method improves the classical DeepLabV3+ network structure based on (1) replacing the backbone network using MoblieNetV2; (2) adjusting the atrous spatial pyramid pooling null rate; and (3) adding the attention mechanism and the convolutional block attention module. Experiments were conducted with self-constructed sample datasets, where the method was compared and analyzed with a fully convolutional network (FCN) and U-Net and ShuffleNetV2 networks; the migration of the method was tested as well. The results show that the method in this paper is better than FCN, U-Net, and ShuffleNetV2, and reaches 92.27%, 91.48%, and 85.63% on the accuracy evaluation indices of overall accuracy, MarcoF1, and mean intersection over union, respectively. Furthermore, the segmentation results are accurate and complete, which effectively alleviates misclassifications and omissions of urban vegetation; moreover, it has a certain migration ability that can quickly and accurately classify the vegetation.

**Keywords:** urban vegetation; unmanned aerial vehicle; feature engineering; deep learning; feature optimization; attention mechanism; DeepLabv3+

## 1. Introduction

Urban vegetation is a key component of urban ecosystems, influencing the urban landscape pattern; it has functions such as absorbing noise, reducing haze, and mitigating the urban heat island effect [1–3]. Classifying and extracting urban vegetation information have important research significance and application value in various fields such as urban land use change, ecological environment monitoring, urban vegetation monitoring, and urban planning [4,5].

Traditional vegetation classification is mainly based on field surveys, which are inefficient for complex terrain and large areas. With the development of remote sensing technology, remote sensing images have become widely used in vegetation survey work [6]. Satellite remote sensing is widely used in large-scale vegetation monitoring due to its advantages of large-scale coverage and long-time sequence observation. The spatial resolution of satellite remote sensing is usually limited by technology and cost; thus, although some

satellites are able to provide high resolution in urban environments, fine vegetation or features are likely to be inaccurately captured, and satellite observations are often affected by cloud cover. In contrast, unmanned aerial vehicle (UAV) remote sensing, characterized by ultra-high resolution and flexibility, is gradually becoming an important tool for urban vegetation surveys [7]. The orthophoto generated by the UAV after processing can extract detailed spatial and textural vegetation information, which is more suitable for the fine classification of urban vegetation [8].

Traditional remote sensing classification methods for vegetation include two categories: pixel-based and object-based. The pixel-based method uses the pixel of the remote sensing image as the smallest classification unit, and uses the feature information in the pixel to judge the vegetation category. This method is mostly used in medium-/low-resolution images, but an image element may contain multiple feature types in high-resolution images. The method does not take into account the up and down information nor the features of the surrounding pixels; thus, there is the phenomenon of "the same object with different spectra, the same spectrum with different objects" [9]. The object-based classification method segments the image into objects with semantic features, and classifies them as the basic classification unit, which comprehensively considers spectral, shape, textural, contextual, and other information aspects; this method has better applications in high-resolution remote sensing vegetation classification [10–12]. However, object-based methods are very much influenced by preset parameters, and improper parameter selection will affect its classification accuracy; furthermore, object-based methods also have problems related to over- and under-segmentation [13].

In recent years, with the rapid development of artificial intelligence technology, scholars have begun using deep learning methods for the remote sensing classification of vegetation [14,15]. Among the deep learning methods, a convolutional neural network (CNN) was first widely used for the remote sensing classification of vegetation. A CNN can automatically extract and learn the vegetation features in an image through convolutional and pooling layers, and simplify the whole classification process by optimizing directly from the original input to the final output in an end-to-end manner. However, the fully connected layer in the CNN structure connects all the neurons in the previous layer and all the neurons in the current layer, resulting in a very large number of parameters. Hence, CNNs are not suitable for pixel-level classification tasks [16,17]. Long et al. [18] proposed a fully convolutional network (FCN) for pixel-level segmentation, which improves the model's ability to perceive features within an image using standard convolutional layers instead of fully connected layers in the CNN and introduces multi-scale feature maps. Since then, scholars have mostly used improved semantic segmentation networks to classify urban vegetation. Xu Zhiyu et al. [19] utilized an improved U-Net to classify urban vegetation into evergreen trees, deciduous trees, and grasslands, with an overall accuracy of 92.73%. Kuaiyu et al. [20] designed a multi-scale feature-aware network to extract and classify urban vegetation in combination with UAV images, with an average overall accuracy of 89.54%. Lin Na [21] et al. proposed a Sep-UNet semantic segmentation model to extract vegetation information in multiple urban scenes, and obtained better results.

Although all of the above networks can effectively classify urban vegetation, the model assigns larger weights to pixels at the edges of different vegetation types during segmentation, resulting in lower edge segmentation accuracy for neighboring samples. To solve this problem, the Google team proposed the DeepLab series of image segmentation networks [22]. This series of networks continues the full convolution operation taken by the FCN, optimizes and improves it, and has been widely used in image processing tasks in recent years. Among them, DeepLabV3+ is the latest improvement in this series of networks, which combines the advantages of encoder–decoder architecture and atrous spatial pyramid pooling (ASPP) to capture a clear target by gradually recovering the spatial information to capture clear target boundaries [23]. Studies have shown that DeepLabV3+ is suitable for the extraction of green space or vegetation information in cities, e.g., Wenya Liu et al. [24] realized the high-precision and high-efficiency automatic extraction of urban

green space through a DeepLabV3+ network. However, the conventional DeepLabV3+ model still suffers from the problems of unrefined classification, large numbers of network parameters, and long training times in urban vegetation classification [25]. Currently, some scholars have tried to improve the DeepLabV3+ network with a lightweight approach, and implemented urban vegetation classification for UAV images [26]. However, the feature learning capability of deep learning models relies on a large amount of training data, and the above scholars only used visible band images as the data; the number of features that the model can learn from the samples is small, thus limiting the performance of the network [27]. In order to overcome this limitation, adding more remote sensing feature data can be considered to make up for the shortcomings of insufficient information from visible light images [28].

In the early stage, due to the small assumption space of shallow machine learning algorithms, it is not possible to express precise mathematical formulas for some complex problems [29]. For this reason, scholars carry out a series of computational processes on the original data through the construction of feature engineering, and refine the more efficient features to facilitate the model's learning, improving its accuracy [30]. Some studies have shown that feature engineering is not only limited to improving the accuracy of shallow machine learning algorithms, but also constructing good feature engineering, which can improve the learning efficiency and classification accuracy of deep learning models. Sun et al. [31] demonstrated that combining the digital surface model with an FCN can improve the ability to semantically segment remote sensing images and significantly improve the classification results. Lin Yi et al. [32] constructed a feature space containing spectral, textural, and spatial information, which effectively improved the fine classification accuracy of urban vegetation. Cui Bingge et al. [33] improved the information extraction accuracy of wetland vegetation by adding a vegetation index to the deep semantic segmentation network. Therefore, the introduction of feature engineering to improve the accuracy of deep learning networks in urban vegetation classification has certain research significance.

Based on the above discussion, this research proposes a UAV remote sensing urban vegetation classification method that combines feature engineering with improved DeepLabV3+. Feature engineering containing vegetation indices and textural features was constructed under feature optimization to increase the number of features in the samples, and the DeepLabV3+ network was improved to increase the classification accuracy and efficiency of the model. Experiments were conducted in several areas of Zunyi City as the study area, using self-constructed sample data to achieve the accurate and complete classification of trees, shrubs, mixed tree-shrubs, natural grasslands, and artificial grasslands.

## 2. Materials and Methods

### 2.1. Data Source and Study Areas

The remote sensing data used in the experiment were taken by a UAV modeled as "Halo PRO", with a maximum take-off mass of 2 kg, a cruising speed of 70 km/h, a fuselage length of 1.5 m, a wing length of 2 m, and a catapult launching mode for take-off. The drone was equipped with a "SNOY A5100" camera with a resolution of 6000 × 4000 and three RGB bands. The time of aerial recovery was in autumn when the weather was clear, and with little wind and stable airflow. The UAV was set to fly at a height of 960 m, and the overlap rate between the heading and the side direction was 80%. The flights were planned using the Pix4D capture mission planner. The orthophoto was obtained after splicing and processing with the Pix4D, and the resolution of the image was better than 0.2 m. The CGCS2000 (EPSG:4490) coordinate system was used.

Zunyi City is located in the northeastern part of the Yunnan–Guizhou Plateau and the northern part of Guizhou Province, with a subtropical monsoon humid climate and complex vegetation types. In this research, five areas with high vegetation coverage in the built-up area of Zunyi City were selected as the study areas, i.e., study areas A to E in Figure 1. Study areas A (area: 80.8 ha), B (area: 200.0 ha), and C (area: 28.5 ha)

contain residential areas, schools, factories, squares, parks and commercial centers, etc., which have a high degree of vegetation coverage and are rich in species to meet the needs for model training, and therefore were used as the training areas for model training and validation. Study area D (area: 100.2 ha) is basically the same as the training area in terms of its coverage of vegetation and features, so it was used as the test data to evaluate the classification accuracy. Study area E (area: 49.1 ha) belongs to the old city; the vegetation coverage is more different, and was used for the model migration test.

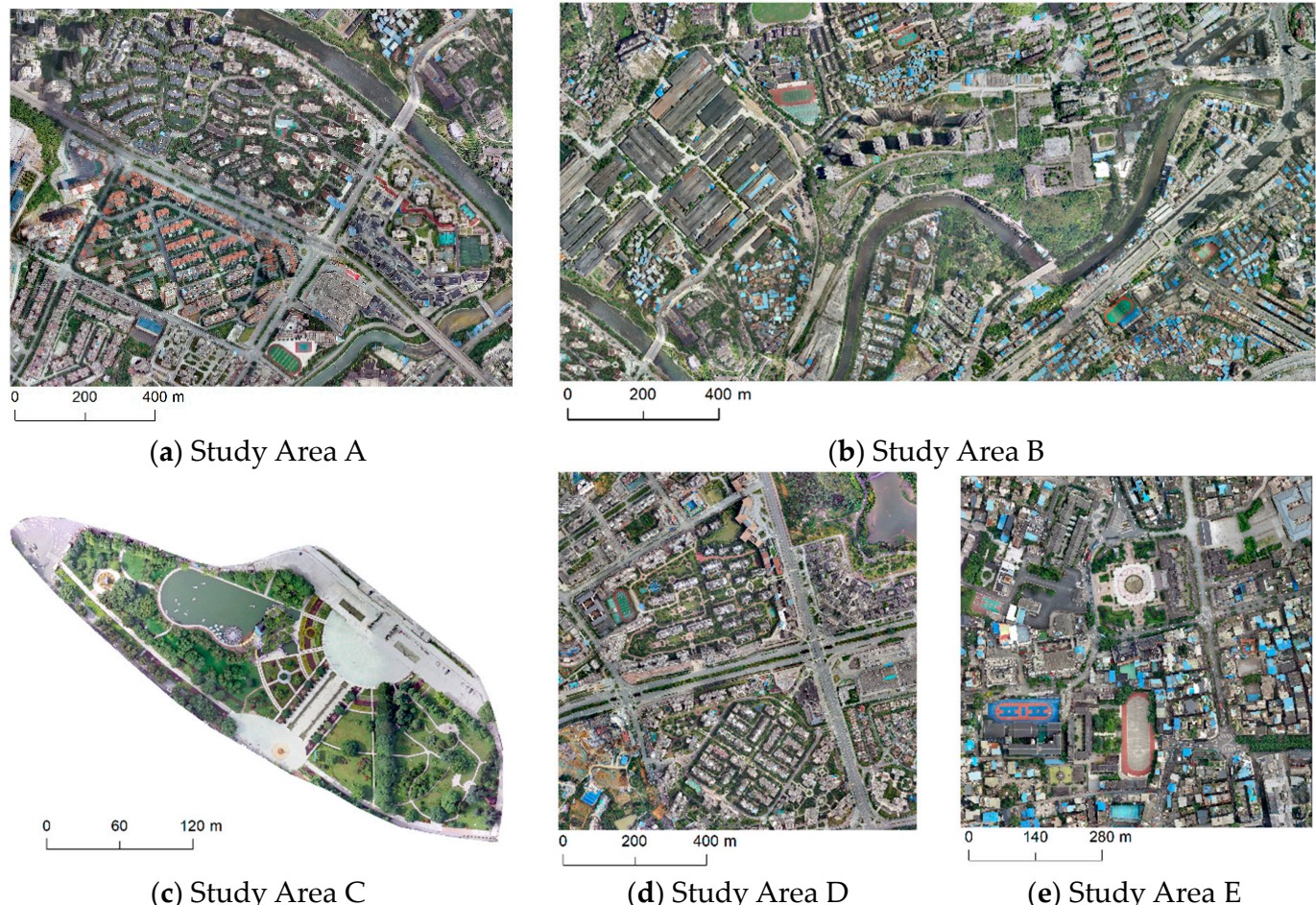

(**a**) Study Area A

(**b**) Study Area B

(**c**) Study Area C

(**d**) Study Area D

(**e**) Study Area E

**Figure 1.** Images of the study area.

*2.2. Methodology*

2.2.1. Technical Workflow

This research proposes an intelligent classification method for urban vegetation, and its flowchart is shown in Figure 2. Firstly, the collected data were preprocessed to generate a UAV mosaic, and the feature engineering is stacking by this image data and remote sensing feature screening algorithm. Then, the urban vegetation sample dataset was constructed by combining the UAV remote sensing image data with the feature engineering. Subsequently, improvements based on the DeepLabV3+ framework and the urban vegetation sample dataset were used to train the improved DeepLabV3+ model. During the training process, the parameters were adjusted according to the convergence of the loss functions of the training samples and the validation samples. The optimal model was selected for accuracy evaluation on the test set, and the model's performance is analyzed via the classification effect diagram, finally realizing a precise and accurate classification method for the UAV.

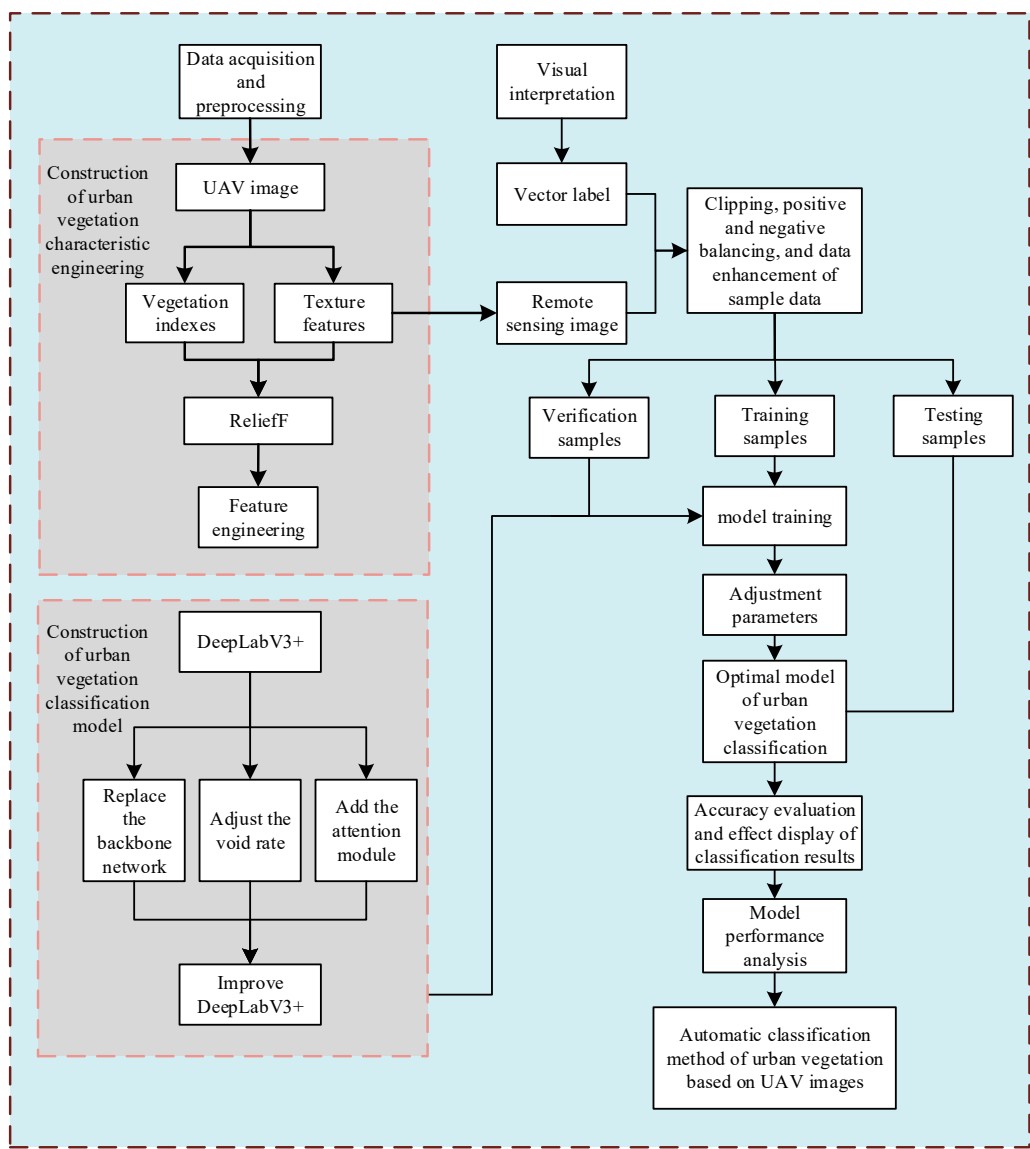

**Figure 2.** Technical flow chart.

### 2.2.2. DeepLabV3+ Network

DeeplabV3+ is a deep learning network dedicated to the task of semantically segmenting images, and the overall structure of the network is shown in Figure 3. DeepLabV3+ continues the structure of the encoder and decoder, but optimizes the network structure. Xception and the ASPP module with atrous convolution were introduced into the encoder of this network to extract image features [34]. In this, null convolution inserts null values at intervals inside the convolution kernel through a preset null rate to expand the receptive field, and the high-level features of the image are obtained by performing different scales of pooling operations on the image features through the ASPP module. Subsequently, these high-level features enter into the decoder after up-sampling, and are fused with the low-level features to form the final feature representation. Finally, the final segmentation result is obtained by reducing the image size through up-sampling [35]. The structure under such optimization can capture contextual information at different scales in the image, enhancing the model's ability to understand local and global information [36], and at the same time mitigate the problem of feature loss, which is beneficial in dealing with complex feature information in urban senses.

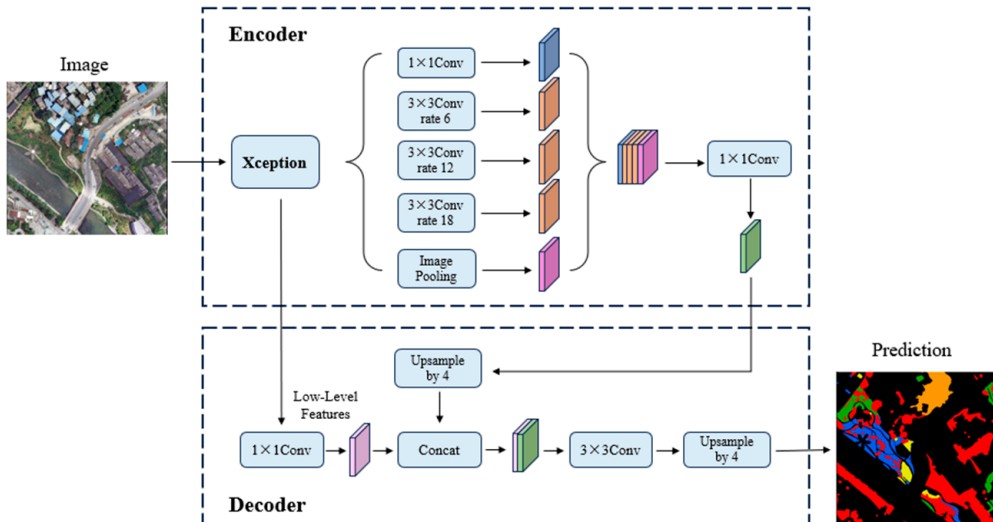

**Figure 3.** Structure of the DeepLabV3+ network.

### 2.2.3. Feature Engineering

Feature engineering is one of the steps in the conventional machine learning. Features are the observations or characteristics on which a model is built, and the process of deriving a new abstract feature based on the given data is broadly referred to as feature engineering [37]. Constructing good feature engineering is the key to extracting high-precision information from remote sensing images [38]. Introducing a vegetation index into deep learning model training can help to better capture the spectral features of vegetation [39] and provide good differentiation between different types of vegetation and covered features; meanwhile, textural features can help the deep learning model better adapt to a variety of different textures and structures in urban environments, making the model more robust. Therefore, combining vegetation indices and textural features can provide multi-source information and help improve the model's urban vegetation classification accuracy. In this study, to characterize different types of urban vegetation, we used ENVI5.3 software to construct the urban vegetation feature project of UAV remote sensing images from two aspects: vegetation features and textural features.

Since the UAV image data used in the experiment contain only three RGB bands and no near-infrared bands, it is impossible to calculate the normalized vegetation index (NDVI); thus, the following vegetation indices that can be calculated with visible light were selected: the visible-band difference vegetation index (VDVI) [40]; the red–green ratio index (RGRI) [41]; the excess green (EXG) [42], excess red (EXR) [43], and excess green–red difference (EXGR) [44] indices; the normalized green–blue difference index (NGBDI) [45]; and the normalized green–red difference index (NGRDI) [46], for a total of seven visible light vegetation indices. The calculation formula is shown in Table 1, where R, G, and B represent the red, green, and blue light bands of the UAV image, respectively.

**Table 1.** Calculation formula for the visible vegetation index.

| Vegetation Index | Calculation Formula |
|:---:|:---:|
| VDVI | $\frac{2G-R-B}{2G+R+B}$ |
| RGRI | $\frac{R}{G}$ |
| EXG | $2G - R - B$ |
| EXR | $1.4R - G$ |
| EXGR | $3G - 2.4R - B$ |
| SNGBDI | $\frac{G-B}{G+B}$ |
| NGRDI | $\frac{G-R}{G+R}$ |

Textural information can provide information about the spatial structure and distribution of feature cover in cities, which can help improve the classification accuracy of models [47]. The biggest advantage of ultra-high-resolution UAV images is the clarity of their textural features. There are large differences in the textures of different vegetation types, and these differences make texture an important feature for classification. Therefore, in this research, textural information was extracted using the gray level co-occurrence matrix (GLCM), whose proponent Haralick further defined 14 textural feature parameters [48], and 8 of the most commonly used textural features were selected in this study: the mean, variance, homogeneity, contrast, dissimilarity, entropy, second moment, and correlation. Eight textural features were extracted using each of the three bands of red, green, and blue light, yielding a total of twenty-four textural features.

### 2.2.4. ReliefF

In this research, 7 vegetation indices and 24 textural features were obtained, but during deep learning model training, a large amount of feature information increases the complexity of training as well as the amount of data to be computed [49], which increases the risk of model overfitting, so optimal feature selection is needed. Feature selection can help to reduce redundant information and identify the most discriminative features for the classification task, which can help to increase the efficiency and performance of the model, as well as improve the accuracy of the model in classifying vegetation in urban environments [50,51]. ReliefF is a filtered feature selection algorithm for remote sensing imagery multi-category problems [52]. Given a sample K, a sample S is randomly selected out of K and initializes the number of iterations m with the number of selected nearest-neighbor samples k. Then, k nearest-neighbor samples P that are of the same class as S are selected from the sample K. Then, k nearest-neighbor samples Q that are not of the same class as S are selected from K. If the distance between sample P and sample S on a randomly selected feature is greater than that between sample Q and sample S, the weight of the feature is decreased, and the opposite is increased. The above process is repeated m times and the results are averaged to obtain the value of each feature parameter weight, calculated as shown in Equation (1):

$$W(A) = W(A) - \sum_{j=1}^{k} \frac{diff(A, S, H_j)}{mk} + \sum_{C \neq class(S)} \frac{\frac{p(C)}{1-p(class(S))} \sum_{j=1}^{k} diff(A, S, M_j(C))}{mk} \quad (1)$$

In Equation (1), $A$ denotes the feature; $W(A)$ denotes the weight of feature $A$; $m$ is the number of repetitions; $k$ is the number of nearest-neighbor samples; $M_j(C)$ is the $j$th nearest-neighbor sample in category $C$; and $diff(A, S_1, S_2)$ denotes the difference between sample $S_1$ and sample $S_2$ on feature $A$, which is calculated via the following formula:

$$diff(A, S_1, S_2) = \begin{cases} \frac{|S_1[A] - S_2[A]|}{\max(A) - \min(A)}, & A \text{ continuous} \\ 0, & A \text{ discrete and } S_1[A] = S_2[A] \\ 1, & A \text{ discrete and } S_1[A] \neq S_2[A] \end{cases} \quad (2)$$

In this research, the weighted values of the vegetation index and textural features are calculated with ReliefF, sorted according to the weighted values, excluded from irrelevant features, and selected as the most relevant features for urban vegetation classification experiments in order to improve the classification accuracy.

### 2.3. Improvements to the DeepLabV3+ Network

Although the DeepLabV3+ network has high edge segmentation accuracy and the ability to integrate multi-scale information, the structure of its network is more complex and requires a large amount of computational resources; this restricts its efficiency in performing fine feature classification. In order to achieve the high-precision and high-efficiency remote sensing classification of urban vegetation, this research improved and

optimized the DeepLabV3+ network structure, used MobileNetV2 as the backbone feature extraction network and modified the null rate of ASPP, and added the attention mechanism convolutional block attention module (CBAM) in the decoding stage. Figure 4 shows the improved network structure.

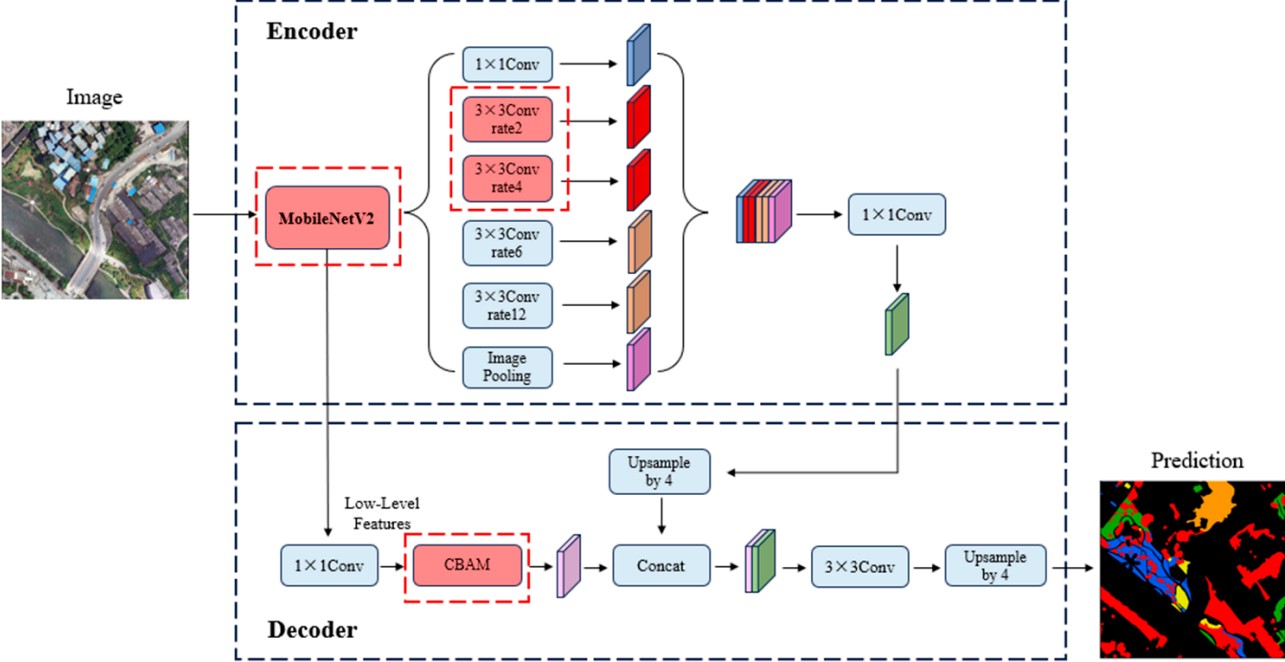

**Figure 4.** Improved network structure of DeepLabV3+ network. The three red dotted boxes indicate model improvements.

### 2.3.1. Replace the Backbone Network

The DeepLabV3+ network usually uses Xpection as the backbone network, which leads to the disadvantages of complex network structure, a large number of parameters, long training times, etc. MobileNetV2 is a lightweight convolutional neural network proposed by Google for mobile devices or embedded systems, which has the advantages of small numbers of parameters, fast speed, and moderate depth [53]; its structure is shown in Figure 5. MobileNetV2 has three important structures: depthwise separable convolution, inverted residuals, and a linear bottleneck. Among them, depthwise separable convolution reduces computation by splitting the convolution operation into two steps, depthwise convolution and pointwise convolution, which reduce computation by several times compared to the standard convolution operation with the same number of weights [54]; inverted residuals reduce computation using $1 \times 1$ convolution before the $3 \times 3$ convolutional layer, and then after the $3 \times 3$ convolution layer, the $1 \times 1$ convolution is used for dimensionality reduction, which allows the network to perform feature expansion and then feature compression; the linear bottleneck structure avoids the activation function from destroying the features, i.e., instead of using the activation function layer for dimensionality reduction, and performs the addition of the residual network directly. MobileNetV2 can provide better feature compression and feature expansion in a lightweight way. At the same time, it can provide better feature representation capability and computational efficiency, so it is often used as the backbone feature extraction network of semantic segmentation models to reduce the number of total parameters of the models and improve their training speeds [55]. Therefore, MobileNetV2 was selected to replace Xpection as the backbone feature extraction network in this research.

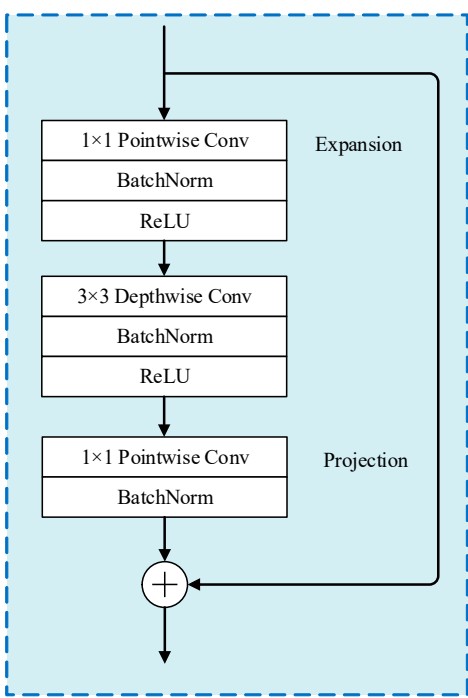

**Figure 5.** Network structure of MoblieNetV2.

### 2.3.2. Adjust the Void Rate

Urban vegetation cover is characterized by irregular edges, complex shapes, different sizes, scattered distributions, etc. [56]. The ASPP used by DeepLabV3+ in the encoder can effectively integrate the contextual semantic information learned by the model, but a hollow convolution rate of 6, 12, 18 is usually used in the ASPP due to the higher spatial resolution of the urban images acquired by UAVs and the clearer boundaries of the features. The overly large cavity rate will lead to the phenomenon that the convolution kernel produces a loss in detail information when performing the operation, and thus the segmentation effect of the boundary becomes worse [57]. Therefore, in this study, the structure of ASPP is adjusted, as shown in Figure 6. Two $3 \times 3$ convolutions with void rates of 2 and 4 are added before the convolution with a void rate of 6, and the convolution with a void rate of 18 is removed, which improves the segmentation ability of the model under high-resolution features.

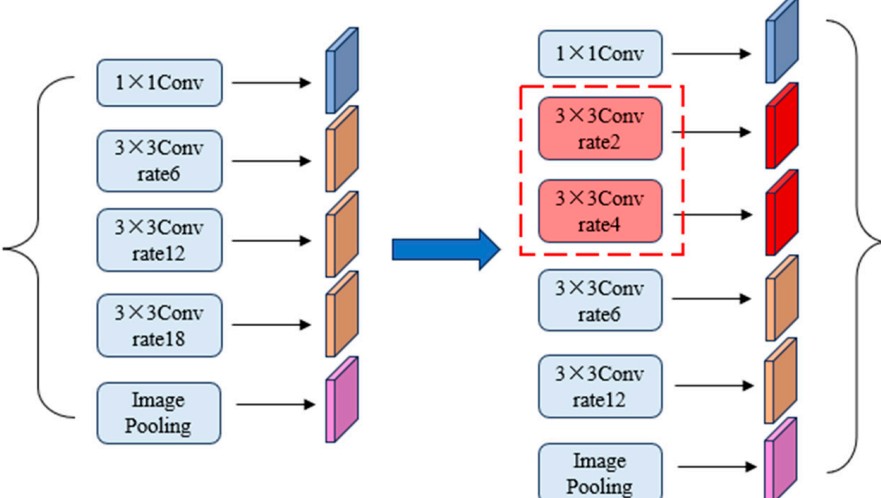

**Figure 6.** Adjustment of the ASPP structure.

### 2.3.3. Adding the CBAM

CBAM (convolutional block attention module) is an attention mechanism module used in deep learning to enhance the attention of convolutional neural networks to important features, and its structure is shown in Figure 7. The CBAM consists of two sub-modules, namely the channel attention module and spatial attention module. Among them, the channel attention module aims to capture the relationships between different channels and weight the features of each channel to better focus on the important channels in the task, while the spatial attention module aims to capture the relationships between different locations in the feature map to better focus on important spatial regions, which helps the module to better focus on spatially influential regions in the task [58]. The CBAM is designed to capture the relationships between different channels and weight the features of each channel by combining the outputs of the channel and spatial attention module's outputs for element-by-element multiplication, thus allowing for attention to both the relationships between channels and important spatial regions in the feature map. This combined attention mechanism helps to improve the network's ability to perceive important features, thus improving the performance of the model [59]. Therefore, in this research, the CBAM was embedded into the decoding stage of DeepLabV3+ to pay attention to vegetation features from both the spatial and channel dimensions, which strengthens the ability to filter non-vegetation features and enhances the feature extraction ability of the model, improving its classification accuracy.

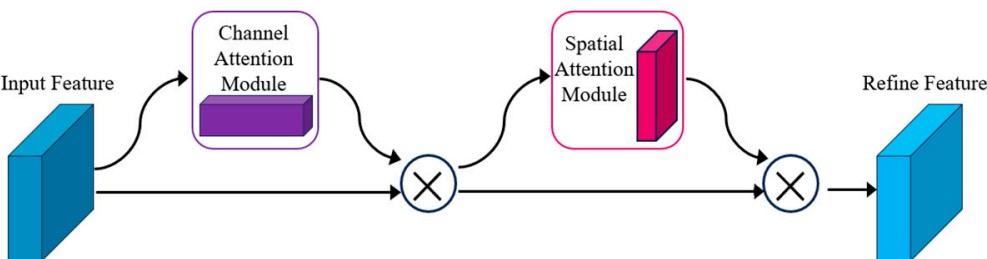

**Figure 7.** Structure of the CBAM.

## 3. Experiments

### 3.1. Constructing the Sample Dataset

The vector label data used in the experiments were all constructed using visual interpretation, i.e., combining the UAV remote sensing image data to classify the vegetation in the training area into five categories, trees, shrubs, mixed trees and shrubs, natural grassland, and artificial grassland, and fully categorizing the non-vegetation features as background values, such as buildings, roads, and water bodies.

In order to reasonably utilize the computer memory, the remote sensing data and labeled data were simultaneously cut into 256 × 256-pixel sample slices using a sliding cut with a 10% overlap rate. It has been shown that the balance of positive and negative samples can improve the performance of the model [60]. For this study, the positive samples are the vegetation samples after being multiclassified, with image element values from 1 to 5, and the negative samples are the other features except the vegetation samples, with an image element value of 0 [61]. Following the principle of selecting high-quality samples, samples with 0-value image elements accounting for more than 80% of the total number of single-sample image elements were removed using histograms; moreover, samples with a certain image element value accounting for 100% of the total number of single-sample image elements were removed from the positive samples in order to balance the multicategory sample size. In addition, in order to keep the training samples sufficient, data augmentation was used for sample expansion, and horizontal flipping, vertical flipping, rotating 90°, rotating 270°, and diagonal mirroring were performed on a sample-by-sample basis. Finally, 11,478 image sample slices and label sample slices each were obtained in the

training region, of which 80% were randomly assigned for the training set and 20% for the validation set. Some of the sample slices are demonstrated in Figure 8.

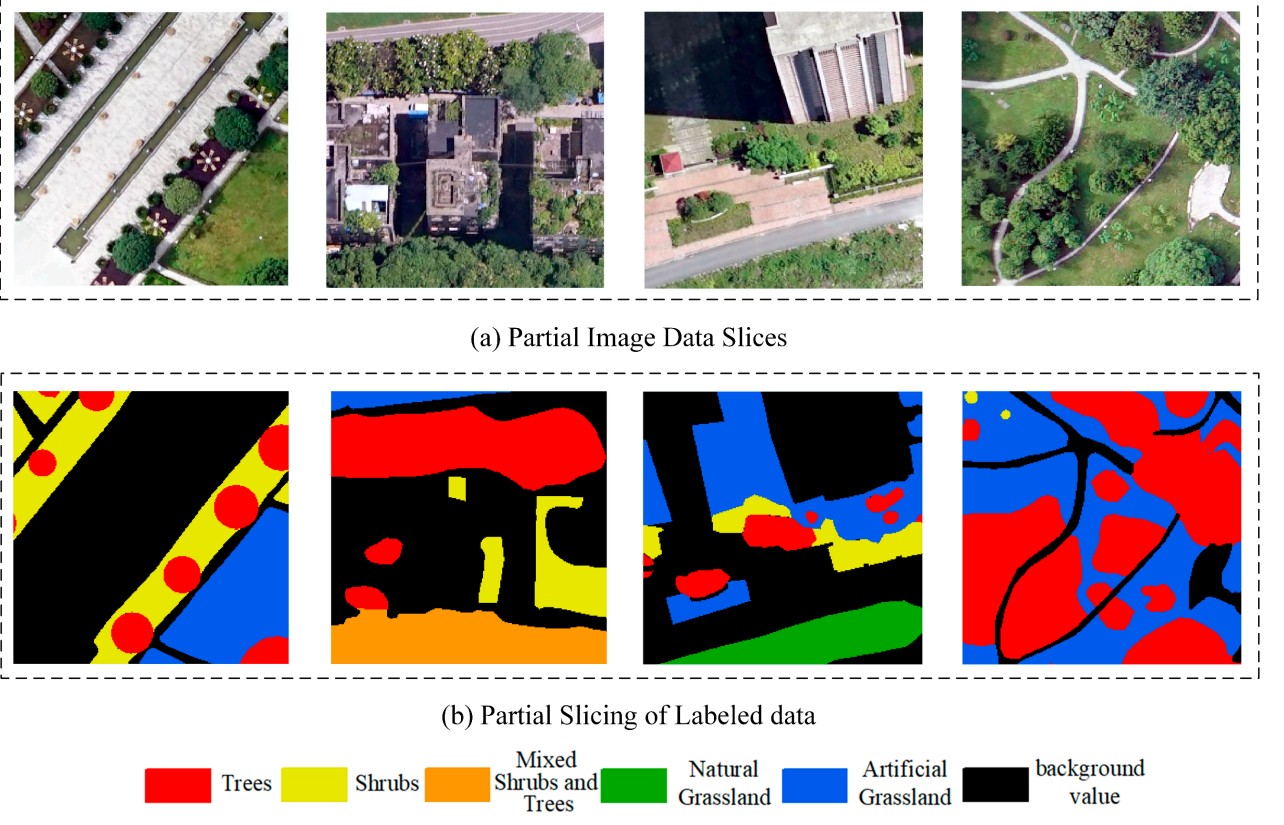

(a) Partial Image Data Slices

(b) Partial Slicing of Labeled data

| | Trees | | Shrubs | | Mixed Shrubs and Trees | | Natural Grassland | | Artificial Grassland | | background value |

**Figure 8.** Display of data slices for some samples.

### 3.2. Feature Optimization

The ReliefF algorithm is executed on the PyCharm platform. Random points are generated on each feature image (vegetation index and textural feature maps), and the gray value extracted from each point is the sample in ReliefF. First, the data are normalized to ensure that the scale of each feature is consistent. For each sample, a weight vector is initialized. For each sample, one of them is randomly selected and the distance between that sample and the others is calculated using the Euclidean distance to find the nearest-neighbor samples of the same kind (i.e., samples belonging to the same category as the current sample) and the nearest-neighbor samples of the dissimilar kind (i.e., samples belonging to a different category than the current sample), respectively. Next, the weights are updated according to the difference with the same class sample and the dissimilar sample, and if the eigenvalue has a greater difference between the same class sample and the dissimilar sample, then its weight will be greater. The above process is iterated several times until the algorithm converges. Finally, the features are ranked according to their final weights, and features with higher weights are ranked higher.

The weights of 7 vegetation indices are calculated first, and then the weights are sorted in ascending order to obtain Table 2. Then, the weights of 24 textural features are calculated, and the weights of the textural features are sorted in ascending order to obtain Table 3. According to the table, the VDVI index contributes the most to the classification of the urban vegetation among the vegetation indices, and the entropy calculated by the green light band contributes the most to the classification of urban vegetation among the textural features, indicating that the VDVI and G_Entropy contribute the most to the classification of urban vegetation in both vegetation and texture, respectively. The textural feature that contributes most to urban vegetation classification is the entropy calculated from

the green light band, indicating that VDVI and G_Entropy have the greatest influence on urban vegetation classification in terms of vegetation and texture, respectively. Therefore, the VDVI index and G_Entropy were selected to construct the feature engineering for urban vegetation classification, which was fused with the sample set of remote sensing data to construct the sample dataset, combining the feature engineering and input into the improved DeepLabV3+ model for training. Table 2 shows the ranking of vegetation index weights.

**Table 2.** Ranking of vegetation index weights.

| Sort | 1 | 2 | 3 | 4 | 5 | 6 | 7 |
|---|---|---|---|---|---|---|---|
| **Vegetation indices** | VDVI | RGRI | NGBDI | ExG | ExGR | NGRDI | ExR |

**Table 3.** Ranking of textural feature weights.

| Sort | Textural Features | Sort | Textural Features | Sort | Textural Features |
|---|---|---|---|---|---|
| 1 | G_Entropy | 9 | R_Correlation | 17 | R_Variance |
| 2 | R_Entropy | 10 | G_Contrast | 18 | B_Variance |
| 3 | R_SecondMoment | 11 | B_Mean | 19 | G_Variance |
| 4 | G_Correlation | 12 | G_Mean | 20 | B_Homogeneity |
| 5 | B_Contrast | 13 | R_Dissimilarity | 21 | B_Correlation |
| 6 | R_Homogeneity | 14 | G_Dissimilarity | 22 | B_Dissimilarity |
| 7 | R_Contrast | 15 | G_Homogeneity | 23 | B_Entropy |
| 8 | R_Mean | 16 | G_SecondMoment | 24 | B_SecondMoment |

R_ stands for calculated from the red band; G_ stands for calculated from the green band; B_ stands for calculated from the blue band.

### 3.3. Experimental Environment and Model Training

The experiments were conducted on a 64-bit operating system of Windows 10, with Tensorflow2.9+Keras as the deep learning framework, and the programming language used was Python3.9. The hardware configuration GPU model is NVIDIA RTX 4090 with 24 GB of video memory, and the CPU model is i9-12900k, with 24 GB of RAM. All deep learning models in this study were built using ReLu (rectified linear unit) as the activation function, and He_Normal as the weight initializer, with appropriate dropout layers added to reduce model overfitting. The hyperparameters of the model in this research were kept the same in training, i.e., the batch size is 32, the number of trainings is 200 epochs, the number of model channels is 6, cross entropy is used as the loss function, and adam is used as the gradient descent optimizer. In order to enable the network to converge quickly and effectively during training, the learning rate was set using segment constant decay, and the initial learning rate was set to 0.001; the learning rate was automatically adjusted to decrease by a factor of 10 every 20 rounds [62]. The model will cause memory overflow if the whole test area image is inputted during prediction, so it is necessary to crop the test area image into $256 \times 256$-pixel slices before prediction, so that the model can read and predict it in pieces, and the prediction results are synthesized and then output.

### 3.4. Precision Evaluation

In order to quantify the model's vegetation classification accuracy on the test images, the commonly used accuracy evaluation metrics in semantic segmentation tasks were selected: the overall accuracy (OA), macro average of the F1-score (MacroF1), intersection over union (IOU), and mean intersection over union (MIOU). OA is the ratio of the number of correctly classified pixels to the total number of all pixels in the classification task, which is an overall index for evaluating the classification effect; MarcoF1 is the F1-score of each vegetation category calculated by precision and recall, and then by finding the mean value. MIOU is the average of the summed IOU values of each vegetation category, which is used to evaluate the overall segmentation accuracy of the model in vegetation classification. In

addition, in order to evaluate the efficiency of the deep learning model, the training time is added to the evaluation index, and the fewer the number of model parameters and the shorter the training time, the higher the efficiency of the deep learning model.

## 4. Results and Analysis

### 4.1. Comparison of Different Methods

In order to verify the advantages of this study's method over other deep learning methods in urban vegetation classification, three representative deep learning methods, FCN, ShuffleNetV2, and U-Net, were selected for a comparative analysis with this study's method. FCN is the first semantic segmentation network [63], ShuffleNetV2 is the classical lightweight convolutional network, and U-Net is the most frequently used network in remote sensing vegetation segmentation [64]. The parameters of FCN, ShuffleNet, and U-Net were kept the same in training, i.e., the batch size is 32, the number of trainings is 200 epochs, the learning rate is 0.0001, the number of model channels is 3, cross entropy is used as the loss function, and adam is used as the gradient descent optimizer.

Table 4 shows an accuracy evaluation for the urban vegetation classification results of the four methods. As can be seen from the table, FCN performs the worst among the four methods, and U-Net performs slightly better than FCN and ShuffleNetV2. This is because the network structure of FCN is relatively simple, and the extraction of features and integration of the global ability are poor; meanwhile, U-Net has stronger feature extraction and global integration ability, but requires more memory and computational resources when using high-resolution drone images, and the corresponding model training time is the longest and more likely to overfit. With computational resources, the corresponding model takes the longest time to train and is more prone to overfitting. In addition, in terms of the overall accuracy (OA), the OA of ShuffleNetV2 is only 0.28% lower than that of U-Net, but its MarcoF1 and MIOU are 7.19% and 6.53% lower than that of U-Net, respectively, which indicates that the segmentation effect of U-Net will be better in the case of small differences between the two in the overall accuracy. ShuffleNetV2 is a classical lightweight network, and its advantage is that the number of parameters is small and the training time is short; however, the disadvantage is obvious, which is the learning degree of the features is insufficient, so that the stability of the model and its edge segmentation accuracy are poor in the vegetation classification task. The method in this paper is better than other methods in all indicators, from the comprehensive indicators; the OA, MarcoF1, and MIOU are 3.68%, 4.81%, and 8.46% higher than U-Net, respectively, which is the most effective among all of the methods. Secondly, the training duration of the model is the shortest, and it is also 0.31 h faster compared to the lightweight network ShuffleNetV2. This is because for this study's method, the depth-separable convolution used in the backbone network MobileNetV2 reduces the number of parameters, while the channel attention mechanism in the CBAM enhances attention to important features and improves the efficiency of feature capture; the combination of the two improves the operational efficiency of the model. In addition, from the perspective of IOU indices of five vegetation categories, the segmentation trend of the four methods is the same, i.e., the segmentation accuracy in descending order is trees > mixed trees and shrubs > natural grassland > artificial grassland > shrubs, with this study's method still showing the best performance, achieving 91.83% and 90.22% of the IOU value of the trees and mixed trees and shrubs, respectively. It shows that the method of this paper has the best accuracy, both in terms of comprehensive indices and various types of vegetation segmentation indices.

In order to qualitatively analyze the effect of the four methods for detail extraction, the test area as a whole and the representative detail areas in it are visualized in Figure 9, and the following analysis is based on Figure 9. From the overall effect of the test area, all four methods can obtain the complete boundary of the image, and can clearly identify trees, shrubs, mixed trees and shrubs, natural grassland, and artificial grassland. However, according to the four detail areas shown, FCN, ShuffleNetV2, and U-Net all have misclassifications and omissions. First of all, the boundary segmentation effect of the FCN

and ShuffleNetV2 methods is poor, and FCN has a greater vegetation mis-segmentation phenomenon, and the vegetation is wrongly classified into the background value in all four detail regions in the figure, which leads to the emergence of voids within the patches. ShuffleNetV2, as a lightweight network, has an insufficient ability to perform multi-scale deep feature extraction, and a lot of broken patches appear. In addition, it can be clearly seen from area 2 that there is a serious pretzel phenomenon in its classification results, i.e., multiple broken patches of different types of vegetation are segmented within the same type of vegetation area. In addition, although the edge segmentation effect of U-Net is better than the previous two methods, due to the model's overly strong feature learning ability, it over-segments trees, which account for a relatively large proportion of the image, and misclassifies the other four vegetation categories and background values as trees, obviously confusing the boundary between trees and roads in region 1, confusing the boundary between trees and shrubs in region 2, and misclassifying tree patches within the shrub patches in region 4. In contrast, the completeness and generality of the segmentation results of this research's method are better than the other three methods, and it can clearly delineate the boundaries between vegetation and background values such as water bodies, roads, and buildings, as well as accurately segment different types of vegetation. These improvements are due to the fact that this study's method introduces feature engineering as well as the null pyramid module that reduces the null rate. The feature engineering enriches the feature information of the sample dataset, in which the vegetation index improves the ability of the model to distinguish the vegetation from the background values; moreover, the textural features increase the detail information of the vegetation and improve the ability to distinguish between different types of vegetation, whereas the adjusted nulling pyramid can make full use of the enriched feature information, which improves the model's ability to obtain the contextual information on the high-resolution imagery, fusing the high- and low-dimension features that can compensate for the loss of boundary information during high-level feature extraction, thus improving the segmentation effect of the boundary contours of the five vegetation types.

**Table 4.** Comparison of accuracy of different methods.

| Methods | OA/% | MarcoF1/% | IOU/% | | | | | MIOU/% | Duration of Training/h |
|---|---|---|---|---|---|---|---|---|---|
| | | | Trees | Shrubs | Mixed Shrubs and Trees | Natural Grassland | Artificial Grassland | | |
| FCN | 84.35 | 78.59 | 78.67 | 55.96 | 76.93 | 68.74 | 65.90 | 68.24 | 6.57 |
| ShuffeNetV2 | 87.31 | 79.48 | 80.11 | 59.36 | 78.04 | 71.66 | 64.01 | 70.64 | 3.93 |
| U-Net | 87.59 | 86.67 | 83.64 | 65.32 | 84.82 | 81.36 | 70.73 | 77.17 | 7.32 |
| Method of this study | **92.27** | **91.48** | **91.83** | **74.57** | **90.22** | **89.20** | **82.31** | **85.63** | **3.62** |

In summary, the method in this paper utilizes the improved DeepLabV3+ network and combines this with constructed feature engineering, which can not only effectively extract the detail information of different vegetation types in the city on UAV images, but also accurately obtain the vegetation boundaries as well as effectively reduce the misclassification and omission of urban vegetation.

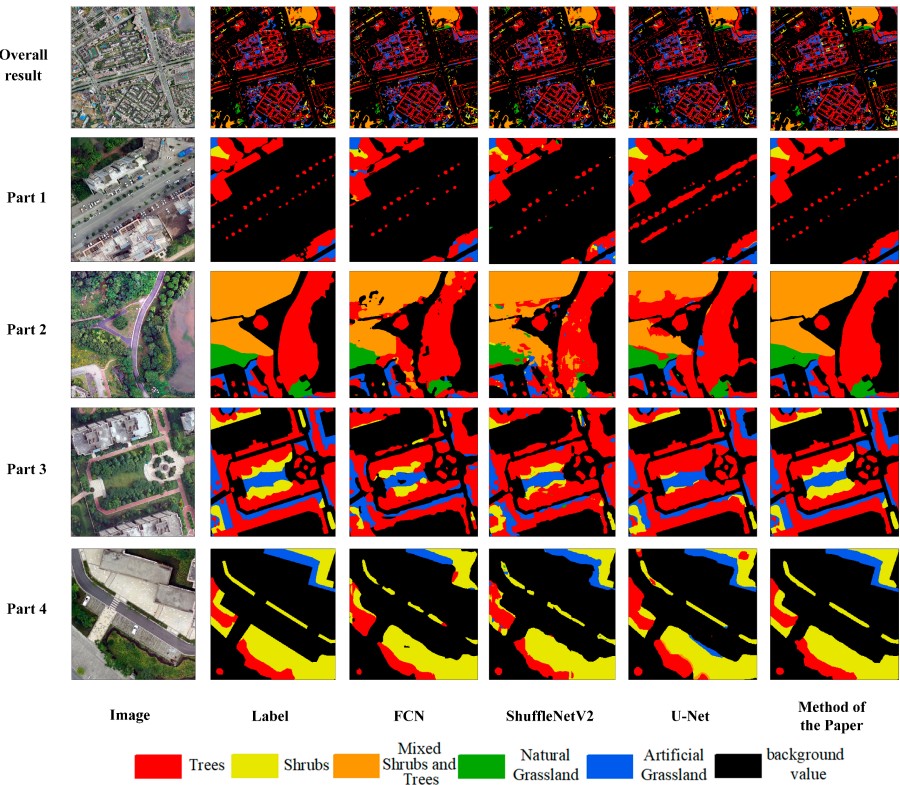

**Figure 9.** Comparison of classification results of different methods in the test area.

### 4.2. Validation of the Effectiveness of the Improvement Mechanism

Layer-by-layer experiments of urban vegetation classification were carried out on UAV images in study area D to further prove the effectiveness of replacing the backbone network, adjusting the null rate of ASPP, adding the attention mechanism module, and combining with feature engineering. Method D1 is based on the conventional DeepLab V3+. Based on the conventional DeepLabV3+, the backbone network Xception is replaced by MobileNetV2 to obtain method D1, and then the void rate of ASPP is adjusted to obtain method D2 based on method D1; then, the CBAM is added in the decoding stage of method D2 to obtain the improved DeepLabV3+ network. Finally, the improved DeepLabV3+ network is combined with feature engineering to obtain the method used in this study.

The results of the accuracy evaluation of the above layer-by-layer experiments are presented in Table 5. As can be seen from the table, compared with the conventional DeepLabV3+, the training time of method D1 is shortened by 1.34 h, and the OA, MarcoF1, and MIOU are improved by 0.82%, 0.49%, and 0.34%, respectively, which proves that the selection of MobileNetV2 as the backbone network can substantially improve the model training efficiency while also improving the classification accuracy. The OA, MarcoF1, and MIOU are improved by 0.53%, 0.85%, and 2.12%, respectively, compared with D1. Method D2's segmentation accuracy is significantly improved, and its training time is further shortened by 0.91 h, which indicates that the ASPP after shrinking the null rate is more suitable for the vegetation classification of high-resolution UAV imagery, and it can effectively improve the model's efficiency. Compared with method D2, the improvement in DeepLabV3+'s OA, MarcoF1, and MIOU is 0.78%, 1.29%, and 2.33%, respectively, which indicates that the addition of the CBAM's attention mechanism can improve the classification accuracy, but because this module introduces a small number of additional parameters, the training time increased by 0.16 h; this study's method combines the feature engineering on the basis of improvements to DeepLabV3+ with its OA, MarcoF1, and MIOU improved by 1.93%, 1.18%, and 4.91%, respectively, indicating that constructing feature engineering can effectively improve vegetation classification of the deep learning model. This especially improves the overall segmentation accuracy. Feature engineering increases the number of

features in the sample so that the model can learn more detailed information; this adds a small time cost to improve the accuracy, so the training time is slightly increased, which also shows the necessity of feature optimization in this research. Only by selecting the features with the highest relevance can the enhancement effect be maximized.

**Table 5.** Results of validation of the effectiveness of the improved method.

| Method | Improved Mechanisms | | | | OA /% | MarcoF1/% | MIOU/% | Duration of Training/h |
| --- | --- | --- | --- | --- | --- | --- | --- | --- |
| | Mobile-NetV2 | Adjustment of ASPP | CBAM | Feature Engineering | | | | |
| DeepLabV3+ | | | | | 88.21 | 87.67 | 75.93 | 5.37 |
| D1 | √ | | | | 89.03 | 88.16 | 76.27 | 4.13 |
| D2 | √ | √ | | | 89.56 | 89.01 | 78.39 | 3.32 |
| Improved DeepLabV3+ | √ | √ | √ | | 90.34 | 90.30 | 80.72 | 3.48 |
| Method of this study | √ | √ | √ | √ | 92.27 | 91.48 | 85.63 | 3.62 |

*4.3. Model Migration Capability Test*

In order to verify the migration ability of this proposed method, the UAV images of study area E were input into the model that was trained for prediction, and the urban vegetation classification results were evaluated using the indicators to obtain the accuracy evaluation shown in Table 6; its classification effect is shown in Figure 10. From Table 6, it can be seen that the method of this study took 40 s in the migration test, and OA and MarcoF1 reached 91.46% and 90.63%, respectively, which shows that this method still maintains high performance and good robustness. In addition, from IOU, MIOU, and Figure 10, this study's method successfully categorized the vegetation in the area into five types of vegetation: trees, shrubs, mixed trees and shrubs, natural grassland, and artificial grassland; furthermore, it shows better segmentation for both overall vegetation and different vegetation. Compared with the test area, the segmentation accuracy of natural grassland is reduced, which may be caused by the low proportion of natural grassland in the image. The above results show that the method of this research achieves high accuracy in the migration test, and the method has a certain migration ability.

**Table 6.** Migration capacity test accuracy.

| OA/% | MarcoF1/% | IOU/% | | | | | MIOU/% | Time Used for Outputting/s |
| --- | --- | --- | --- | --- | --- | --- | --- | --- |
| | | Trees | Shrubs | Time Used for Outputting | Natural Grassland | Artificial Grassland | | |
| 91.46 | 90.63 | 90.45 | 74.35 | 90.13 | 67.96 | 80.19 | 80.62 | 40 |

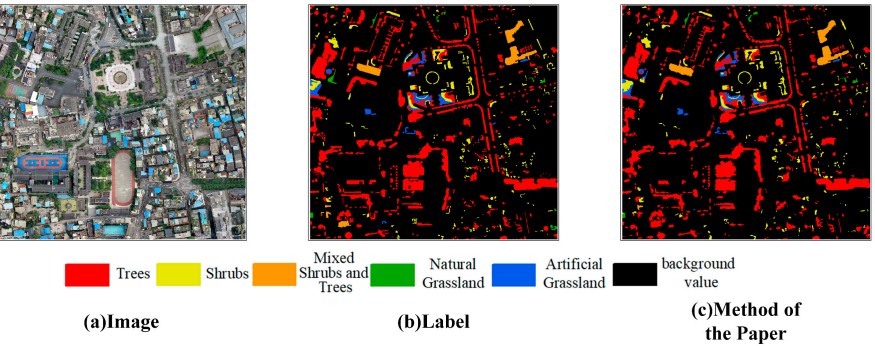

**Figure 10.** Classification effect of migration ability test.

## 5. Discussion

In this study, UAV RGB images were used as the data source. By replacing the backbone network, adjusting the null rate of ASPP, and adding the attention mechanism, we effectively improved the performance of DeepLabv3+. Zhang et al. [26] employed an improved model, akin to our study, to extract vegetation from two residential areas in Nanjing, China, using UAV imagery. The obtained results remain as excellent. The model improvement significantly reduced the training time by 25.70%. In contrast, our method achieves an even more substantial reduction of 32.60%. The key distinction lies in the fact that the former incorporates only channel attention, while we introduce both channel attention and spatial attention. This discrepancy elucidates that CBAM enhances model performance by optimizing the combination of these two attention mechanisms. Lin et al. [32] proposed a methodology involving feature engineering along with an improved deep learning model for classifying vegetation in a plot in Jiaozuo City, Henan Province, China, Binhe Garden District, based on remotely sensed imagery. Due to constraints such as small sample sizes and a lack of migratable sample datasets in their study, OA for vegetation classification was limited to 83.30%, falling short of the OA achieved in our study (92.27%). Our experiment benefits from an ample supply of training samples, facilitating the deep learning model to achieve higher accuracy.

We added feature engineering to improve DeepLabv3+, which further improves the model's classification results for urban vegetation, especially with the MIOU being 4.91% higher than the method without feature engineering. This result is consistent with previous findings on the combination of deep learning and feature engineering for vegetation extraction [65]. Xu et al. [66] added the vegetation index into a deep learning model for urban vegetation remote sensing classification, and also achieved high accuracy. However, the extraction accuracy of grassland in the study is 75%, while the accuracy of both natural and artificial grassland extracted by our method is above 80%. This may be due to the fact that we added more GLCM into our feature engineering. Since the texture of grass is flatter and obviously different from trees and shrubs, the GLCM improves the segmentation accuracy of grass. In addition, the UAV images used in our study were all from the same altitude in the same season. Considering the possible effects of images from different seasons and altitudes on vegetation classification, we will combine the remote sensing data from different shooting altitudes and seasons to assist in the study of vegetation classification in the future. At the same time, we will consider adding more types of feature information to the feature engineering to enhance the confidence of the model's classification decision making, in order to further improve its performance.

## 6. Conclusions

The existing urban vegetation fine classification method requires a lot of time, and is not effective in categorizing the vegetation. Therefore, this research proposed an automatic urban vegetation classification method that combines feature engineering and improved DeepLabV3+ with UAV images as the data source. Through comparison experiments with different methods to validate improvements in effectiveness and the migration test, the following main conclusions are drawn: This research's method can accurately and completely categorize the vegetation into trees, shrubs, tree-shrub mixes, natural grasslands, and artificial grasslands on the UAV images, and the segmentation effect of trees is the best, achieving a segmentation accuracy of 91.83%. Meanwhile, the feature engineering constructed under feature optimization significantly improves the overall segmentation accuracy of the deep learning model. Replacing the backbone network by adjusting the null rate shortens the model training time while improving its segmentation accuracy. After adding the CBAM, the classification accuracy for urban vegetation is further improved. In conclusion, the improvement mechanisms of this study's method are all effective in enhancing urban vegetation classification. In addition, the method in this paper has high classification efficiency and a certain migration ability, which are suitable for rapid investigations of urban area vegetation. Overall, the method proposed in this paper

can quickly and accurately classify urban vegetation on UAV images, which is of great significance for exploring the changes in and applications of vegetation.

**Author Contributions:** Conceptualization, Q.C. and G.Y.; methodology, Q.C.; software, R.W.; validation, M.L. and Y.L.; formal analysis, Q.T.; investigation, Y.L.; resources, G.Y.; data curation, M.L.; writing—original draft preparation, Q.C.; writing—review and editing, Q.C. and M.L.; visualization, Q.T.; supervision, Q.T.; project administration, G.Y.; funding acquisition, P.C. All authors have read and agreed to the published version of the manuscript.

**Funding:** This research was supported by Guizhou Provincial Science and Technology Program Project—Natural Resource Statistics and Asset Value Assessment Based on Remote Sensing Big Data (No.: Qiankehe Major Special Project [2022] 001), and Guizhou Provincial Science and Technology Program Project-Research on Key Technologies for Remote Sensing Monitoring of Natural Resources Assets in Karst Region (No.: Qiankehe Support [2023] General 176).

**Data Availability Statement:** The data presented in this study are available on request from the corresponding author.

**Conflicts of Interest:** The authors declare no conflicts of interest.

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
