# Peer review of "Urban Vegetation Classification for Unmanned Aerial Vehicle Remote Sensing Combining Feature Engineering and Improved DeepLabV3+"

_forests, doi:10.3390/f15020382_

Round 1
Reviewer 1 Report
Comments and Suggestions for Authors
Dear Authors,
Firstly, I would like to express my gratitude for the work carried out by the authors. I also want to thank them for meticulously preparing the text, almost in the form of a user manual. Understanding the article and the conducted studies required considerable time, hence the delay in my evaluation. I believe that the work carried out by the authors has qualities that can serve as a foundation for further studies in both the literature and the practical application. In fact, I found it to be so detailed that it could be presented as a thesis or even more. However, I think that in the presentation of this wealth of information and the tasks undertaken, some areas lack sufficient details or could be improved.
Unfortunately, I noticed that excessively long sentences are used in the text, often with the use of commas and/or semicolons, which disrupt the fluency of the text. Regarding the development of the method, its application in test fields, and its evaluation in comparison with accepted techniques or approaches, I believe there is no scientific deficiency. There are only a couple of points to note. One is the necessity to elaborate on how the weights presented under the "3.2. Feature Preferences" section were obtained. The other concerns the evaluation of accuracies, particularly in the context of using very high-resolution data. While traditional accuracy analyses are essential, all algorithms used perform similar classifications. The differences lie in the boundaries of the classes. At this stage, it would be more accurate to use accuracy analyses that compare differences between boundaries obtained through digitization and those obtained through classification. You can find studies discussing accuracy analyses for object-based techniques in the literature.
Below and in the attached document, you can find some deficiencies I observed in your article.
Specifying the names of the algorithms used in Figure 1 will enable the reader to construct the process steps more easily.
Page 1 Abstract; Even though the reader may know the meanings of abbreviations, explanations should still be provided. (UAV, CBAM, FCN, OA, MIOU) Additionally, all instances of the abbreviation 'ASSP' in the text are incorrect. It should be 'ASPP' (Atrous Spatial Pyramid Pooling).
Page 2, 2nd paragraph; the phenomenon of "the same object with different spectra, the same spectrum with different objects" [9]. " phenomenon [9].
Page 2; There are never-ending sentences. Please simplify and correct them.

As mentioned above, there are some typos and very long sentences in the text. These need to be corrected.
Reviewer 2 Report
Comments and Suggestions for Authors
Review Forests 17 jan 2024
Page 2 paragraph 2, should discuss spectral mixing in a little more detail
Have you thought about negative templates? My colleagues & I did this with our CNN to greatly improve our classification
Baur, Jasper, Gabriel Steinberg, Alex Nikulin, Kenneth Chiu, Timothy S. de Smet. 2021. How to Implement Drones and Machine Learning to Reduce Time, Costs, and Dangers Associated with Landmine Detection. Journal of Conventional Weapons Destruction 25(1):137-145. https://commons.lib.jmu.edu/cisr-journal/vol25/iss1/29/
Page 2 paragraph 3 line 5, typo, need space after period between sentences
Page 2 paragraph 3 line 9, typo, needs punctuation
Page 2 paragraph 3 line 12, typo, need space after period between sentences
Page 3 line 2, maybe instead use semicolon before the word however
Page 3-9 methods: what UAV did you use? What front and sidelap did you use? What camera model? What elevation? Need at least a paragraph on this.
Page 7: what extra channels are you using? Ahh I see it later you are adding vegetation indices, maybe say this earlier here, again what camera did you use? What are the central frequency and bandwidth of your presumably multispectral layers?
Oh I see, just RGB, again what camera? Are you using raw data or lossy jpegs?...
Page 9: Yeah move the UAV paragraph into the methods, is altitude AGL with waypoints to maintain constant AGL? What mission planning software did you use?
Page 9-11: what photogrammetry softeware did you use to create the orthophotos?
Page 10, cutting it up into smaller pieces is smart, same thing my colleagues and I do
Page 11 line 9 awkward ssentence and typos
Page 11, section 32.: could you explain the ranking a little better, because say with a PCA you quantitatively know the amount of variance each PC axis contains
Page 12 line 4 typo no space after period
Page 17: different seasons and altitudes indeed are a consideration
Comments on the Quality of English LanguageFine, they just have a lot of typos, like no space between periods and new sentences, and sentences that start with the lower case, etc....
Reviewer 3 Report
Comments and Suggestions for Authors
In the submitted work for review, the authors focus on classification aimed at extracting objects with active vegetation.
- The authors use excessively long sentences in the paper, resulting in difficulty understanding the text. For example, "The pixel-based method uses the pixel of the remote sensing image as the smallest classification unit, and uses the feature information in the pixel to judge the vegetation category, which is mostly used in low-resolution images. However, in high-resolution images, a pixel may contain more than one feature type, which does not take into account the upper and lower information and features of the surrounding pixels, and thus there exists the phenomenon of 'the same object with different spectra, the same spectrum with different objects' [9]."
- In Table 1, there are no references to the literature.
Author Response
Dear Reviewers:
Thank you very much for taking the time to review this manuscript. These comments are valuable and helpful to us in revising and improving the manuscript, as well as providing important guidance for our research. We have carefully studied these comments and made changes that we hope will be accepted. The revised parts are marked in red in the manuscript. In addition, we have made a few adjustments to the structural order of the manuscript. Major corrections and responses to reviewers' comments are listed below.
Responds to the reviewer’s comments:
Comments 1: The authors use excessively long sentences in the paper, resulting in difficulty understanding the text.
Response 1: We gratefully appreciate for your valuable comment. We have improved the English of the manuscripts.
Comments 2: In Table 1, there are no references to the literature.
Response 2: Thank you for pointing this out. We are very sorry for our careless mistake. Therefore, we have cited the literature [39]-[45] on page 6.
Thank you again for your positive comments and valuable suggestions to improve the quality of our manuscript.
With best regards,
Qianyang Cao
Reviewer 4 Report
Comments and Suggestions for Authors
Thank you for the opportunity to review this interesting article.
This paper addresses the challenges of misclassification and omission in fine classification results of urban vegetation, common in current remote sensing classification methods. The proposed intelligent urban vegetation classification method integrates feature engineering and an enhanced DeepLabV3+ based on UAV visible light images.
In my opinion, the paper is not sufficiently well-structured. The chapters are poorly organized, and the text lacks coherence, making it difficult to form a clear understanding of the steps, algorithms, and their adjustments. Additionally, I would like to draw attention to the lack of literature references (e.g., https://doi.org/10.3390/ijgi12110454; or a comprehensive search on MDPI: https://www.mdpi.com/search?q=DeepLabV3+uav). Nevertheless, the topic of automatic urban vegetation extraction is interesting, and I encourage the authors to improve their work and resubmit.
All my other comments are provided in the accompanying document.
I believe the paper cannot be accepted in its current form and should be rejected.
Thank you.

Reviewer 5 Report
Comments and Suggestions for Authors
Dear Authors,
The manuscript "Urban Vegetation Classification for UAV Remote Sensing Combining Feature Engineering and Improved DeepLabV3+" addresses the detection and classification...
The term mosaic or orthomosaic is more appropriate than orthophotography. The more appropriate term for pixel size (GSD) is spatial or geometric resolution.
In order to help improve the manuscript, I made several comments on the digital file. I would like to draw your attention to the following recommendations:
1) Abstract: revise the confusing wording. Define the problem, the method used and the improvement obtained compared to the usual methods.
2) Page 2: "The orthophoto generated by UAV". Processing usually takes place after the aerial images have been taken.
3) Object-based also has problems related to over and under segmentation. Please add this information.
4) The term feature engineering used is nothing more than information added as input to the network. Is this the most appropriate term? Vegetation index and co-occurrence matrix are more usual terms.
5) Table 1: vegetation indices are usually calculated using reflectance. Specify the model used to convert the gray levels present in the image into reflectance values.
6) Detail the software and hardware used in photogrammetric processing, calculation of vegetation indices and co-occurrence matrices.
7) Present the parameters used for photogrammetric processing.
8) Present the area values (ha) for each site analyzed.
9) Table 2 and 3: present the variance/ranking values that are explained by each variable.
10) Table 4: for each column, highlight the best value in bold.
11) Additionally, you can compare the results obtained in your study with those obtained by the authors of the theoretical framework. This provides more scientific support for the study and improves the discussion of the results.
12) Revise the wording of chapter 3.6.2 as it is very confusing.
13) Revise the conclusions so that they adequately respond to the objectives of the study. It is not necessary to repeat the results obtained in this chapter.
I end my review by congratulating you on your study and the version of the manuscript you have submitted.
Respectfully,

Regarding the writing of the manuscript, check the punctuation, especially the comma. When an acronym is presented for the first time in the text, its meaning should be presented. Avoid using long paragraphs that tend to make reading confusing. I have marked in the comments of the digital file the passages in which the writing should be improved.
Round 2
Reviewer 4 Report
Comments and Suggestions for Authors
Dear authors,
Your paper has been significantly improved. I think there are a few more things that need to be corrected:
The abstract is again written too extensively, it should have a maximum of 200 words.
I did not find the part where the parameters used for each of the models are listed. Therefore, the parameters that were used should be specified for each model.
Some of the figures are of poor quality, they need to be fixed (Figures 8, 9 and 10)
The discussion is short, a comparison of the results with other studies can be added.
Reviewer 5 Report
Comments and Suggestions for Authors
Dear Authors,
The manuscript "Urban Vegetation Classification for UAV Remote Sensing Combining Feature Engineering and Improved DeepLabV3+"presents several changes in the wording and improvements in figures, tables and graphs. The new information added has increased the final quality of the manuscript.
Comparing the second version with the first, I have seen that you have implemented or justified several changes. I would like to draw your attention to these changes:
- Reference the term feature engineering for machine learning,
- The values of the study areas can be approximate and a good alternative for obtaining these values is via Google Earth Pro and
- Analyzing tables 2 and 3, I noticed that rank number eight is missing Please confirm and update the tables if necessary.
Thank you for sending me the cover letter, which helped me with the revision work.
Congratulations.
